🔓 | **Open Peer Review** | Clinical Microbiology | Research Article

# A low-cost culture- and DNA extraction-free method for the molecular detection of pneumococcal carriage in saliva

Chikondi Peno,[1] Tzu-Yi Lin,[1] Maikel S. Hislop,[1] Devyn Yolda-Carr,[1] Katherine Farjado,[1] Anna York,[1] Virginia E. Pitzer,[1] Daniel M. Weinberger,[1] Amy K. Bei,[1] Orchid M. Allicock,[1] Anne L. Wyllie[1]

**ABSTRACT** Molecular methods have improved the sensitivity of the detection of pneumococcal carriage in saliva. However, they typically require sample culture enrichment and nucleic acid extraction prior to performing the detection assay and may limit scalability for extensive surveillance of pneumococcus, particularly in low-resource settings. We evaluated the performance of a DNA-extraction-free method for the detection of pneumococcus in saliva. We developed a streamlined qPCR-based protocol for the detection of pneumococcus, omitting culture enrichment and DNA extraction. Using saliva samples collected from children attending childcare centers (New Haven, CT, USA), we evaluated the detection of pneumococcus using saliva lysates as compared to purified DNA extracted from culture-enriched aliquots of the paired samples using qPCR targeting the pneumococcal *piaB* gene. Of the 759 saliva samples tested from 92 children [median age 3.65 years; IQR (2.46–4.78)], pneumococcus was detected in 358 (47.2%) saliva lysates prepared using the extraction-free protocol and in 369 (48.6%) DNA extracted from culture-enriched samples. We observed near-perfect agreement between the two protocols (Cohen's kappa: 0.92; 95% CI: 0.90–0.95). Despite a high correlation between $C_T$ values generated by the two methods ($r = 0.93$, $P < 0.0001$), the $C_T$ values generated from saliva lysates were higher (lower concentration) than those from culture-enriched samples ($\Delta C_T = 6.69$, $P < 0.00001$). The cost of detecting pneumococcus using saliva lysates was at least fivefold lower (US$2.53) compared to the cost of the culture-enriched method (range: US$13.60–US$19.46). For pneumococcal carriage surveillance in children, our findings suggest that a DNA extraction-free approach may offer a cost-effective alternative to the resource-intensive culture-enrichment method.

**IMPORTANCE** Surveillance for carriage of pneumococcus is a key component of evaluating the performance of pneumococcal vaccines and informing new vaccination strategies. To improve the scalability of pneumococcal carriage surveillance, we show that molecular detection of pneumococcus in saliva from children can be performed without culture enrichment and DNA extraction. Our findings show that using the extraction-free method can improve surveillance efforts for pneumococcal carriage in children, overcoming the resource-intensive hurdle that comes with the use of molecular methods, particularly in low-resource settings.

**KEYWORDS** pneumococcus, saliva, qPCR, carriage

$S$treptococcus pneumoniae (pneumococcus) is a leading cause of bacterial infections including pneumonia, meningitis, bacteremia, and sepsis. Globally, more than 300,000 deaths in children under the age of five are caused by pneumococcal diseases, with the highest burden of disease experienced in low-to-middle-income countries (1). Carriage of pneumococcus in the upper respiratory tract (URT) is common and usually asymptomatic but also a prerequisite for invasive pneumococcal disease and a source

Address correspondence to Chikondi Peno, chikondi.peno@yale.edu, or Anne L. Wyllie, anne.wyllie@yale.edu.

Chikondi Peno and Tzu-Yi Lin contributed equally to this article. Authorship was determined upon seniority in the lab.

D.M.W. has received consulting fees from Pfizer, Merck, and GSK and is a PI on research grants from Pfizer and Merck. A.L.W. has received consulting and/or advisory board fees from Pfizer, Merck, Diasorin, PPS Health, Co-Diagnostics, and Global Diagnostic Systems for work unrelated to this project and is a principal investigator on research grants from Pfizer, Merck, NIH RADx UP, and SalivaDirect, Inc. to Yale University and on research grants from NIH RADx, Balvi.io, and Shield T3 to SalivaDirect, Inc. All other co-authors declare no conflict of interest.

of pneumococcal transmission in the community (2, 3). Following the introduction of pneumococcal vaccines, carriage surveillance has been pivotal for evaluating vaccine performance and informing new vaccination strategies (4, 5). Because pneumococcal vaccines only target a subset of >100 known pneumococcal serotypes (6), carriage surveillance is also crucial for monitoring persisting vaccine serotypes and detecting the emergence and expansion of non-vaccine serotypes that can be targeted in new vaccine formulations (7).

The surveillance of pneumococcal carriage dynamics relies on establishing methods of detection that are sufficiently sensitive across all ages and scalable in different settings, particularly in settings with a high disease burden (8). While the current recommended gold standard method for detecting pneumococcal carriage in the URT is the conventional culture of a nasopharyngeal swab (9), molecular detection methods improve the sensitivity of pneumococcal carriage detection when applied to nasopharyngeal swabs and other URT sample types including oropharyngeal swabs and saliva (10–13). Despite being more sensitive for pneumococcal carriage detection, molecular methods are resource-intensive, requiring extraction of DNA from the sample prior to testing. Moreover, for oral sample types (oropharyngeal swabs and saliva), a culture step using a selective medium is recommended prior to nucleic acid extraction to enrich the sample for pneumococcus to further improve the sensitivity of detection in these polymicrobial samples (14). Together, these requirements may limit the scalability of extensive surveillance efforts, particularly in resource-limited settings. Saliva, however, is an ideal sample for pneumococcal community surveillance studies; it is a non-invasive sample type that can be reliably self-collected (15), negating the need for trained healthcare professionals, typically required for the collection of swabs. Overall, this leads to a lower sample collection burden to study participants (minimizing potential testing aversion), and/or clinical personnel, which is critical particularly in longitudinal studies. Additionally, we have previously demonstrated that encapsulated pneumococci remain viable in raw, unsupplemented saliva with a stable bacterial load seen for 24 hours in the absence of cold-chain transport (i.e., ~19°C–30°C) (16), further alleviating collection and transport burden for studies conducted in remote or resource-limited settings.

During the COVID-19 pandemic, Vogels et al. developed a saliva-based, nucleic acid extraction-free qPCR test for the detection of SARS-CoV-2 with the aim of reducing the burden and cost of molecular detection (17). Replacing nucleic acid extraction with a simple enzymatic lysis or heat treatment step was possible without compromising the accuracy and efficiency of testing for SARS-CoV-2 (17). Here, we evaluated the application of this extraction-free qPCR approach to detect pneumococcus in saliva samples to explore the feasibility of a low-cost, molecular screening method for pneumococcal carriage.

## RESULTS

### Performance of the extraction-free method in the detection of pneumococcal carriage

A total of 759 saliva samples collected from 92 children attending childcare centers (median age: 3.23 years, IQR: 1.8–4.5 years) were used to compare the performance of two pneumococcal detection protocols: a well-established method in which DNA is extracted following sample culture enrichment (CE) (3, 18–20) and an extraction-free method, originally developed and extensively used for the detection of SARS-CoV-2 (Fig. 1) (17). Of the 759 samples, pneumococcus was detected in 378 (49.8%) samples using at least one of the methods, with a near-perfect agreement between the two protocols (Cohen's kappa = 0.92, 95% CI: 0.90–0.95; Table 1). The culture-enrichment protocol detected slightly more positive samples (369/759, 48.6%) compared to the extraction-free method (358/759, 47.1%). We observed discordant results in 29/378 (7.6%) samples, of which 20/29 (68.9%) tested positive using the culture-enrichment method but negative by the extraction-free protocol, while 9/29 (31.0%) tested positive using the extraction-free method but negative following culture enrichment (Fig. 2 ).

Culture enrichment and nucleic acid extraction protocol

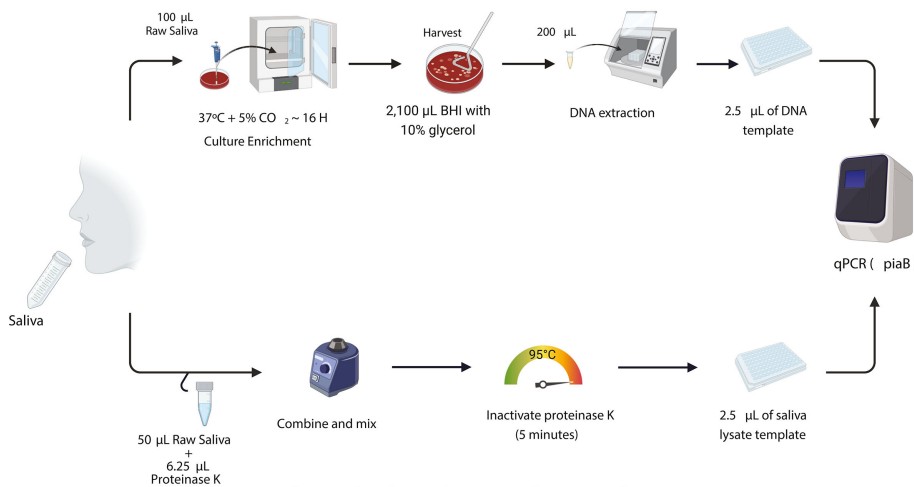

**FIG 1** Schematic overview of the sample preparation methods compared for the detection of pneumococcal carriage in saliva samples. Raw, unsupplemented saliva was tested using qPCR targeting the pneumococcus-specific gene, *piaB*, following culture enrichment on trypticase soy agar supplemented with 5% defibrinated sheep blood and 10 mg/L gentamicin (Gent plates, ThermoFisher Scientific) prior to DNA extraction, and a simplified culture- and extraction-free approach in which saliva was simply mixed with proteinase K and heated at 95°C for 5 minutes. Figure created with Biorender.com.

There was no significant difference in the detection rate of the two methods according to McNemar's test ($P = 0.06$). Almost all samples that yielded discordant results (28/29, 96.5%) had higher qPCR $C_T$ values ($C_T > 30$), with the majority having $C_T$ values $> 35$ (23/29; 79.3%) (Table S1).

The accuracy of detection when testing samples using the extraction-free method was not inferior to that of the culture-enrichment method. When using the culture-enrichment method as the "reference standard," all measures of accuracy including sensitivity (94.7%), specificity (97.7%), positive predictive value (97.5%), and negative predictive value (95%) were greater than 90% (Table 1). Since the culture-enrichment method is not an established reference standard, we further compared the accuracy of the detection methods using a composite reference, where pneumococcal carriage positivity was defined as any sample yielding a $C_T$ value $< 40$ from either the culture-enrichment or the extraction-free method. For both methods, there was a near-perfect agreement with the composite reference (culture enrichment, Cohen's kappa = 0.98 vs extraction-free, Cohen's kappa = 0.95; Table 1). However, there was significant disagreement for both methods when compared to the composite reference (culture enrichment, McNemar's test $P = 0.007$ vs extraction-free, Mcnemar's test $P < 0.0001$; Table 1). Nonetheless, all measures of detection accuracy remained above 90% (Table 1).

The limit of detection of pneumococcus when testing saliva lysates processed by the extraction-free method was determined to be $5 \times 10^2$ CFU/mL (Fig. S1B; Table S2). When testing DNA extracted from culture-enriched samples, we were able to detect pneumococcus down to 8 CFU/mL (Fig. S1C; Table S2), demonstrating the lower quantification capacity of the extraction-free method when compared to the culture-enrichment method.

We observed a high correlation between the $C_T$ values obtained using the extraction-free method and the culture-enrichment method ($r = 0.93$, $P < 0.00001$; Fig. 3A). Among the samples yielding positive results on both methods, lower bacterial quantities (as inferred by higher qPCR $C_T$ values) were observed when using the extraction-free protocol as compared to the culture-enrichment method ($\Delta C_T = 6.69$, $P < 0.00001$; Fig 3B). Similarly, adjusting for differences in sample storage conditions between the sampling periods, we still observed significantly lower bacterial quantities (as inferred

**TABLE 1** Sensitivity and specificity of pneumococcal carriage detection in saliva samples when tested using qPCR following culture-enrichment or extraction-free sample processing methods

| | | Positive | Negative | P-value | Concordance (kappa) | PPV (%) | NPV (%) | Sensitivity (%) | Specificity (%) |
|---|---|---|---|---|---|---|---|---|---|
| **Culture-enriched (reference)** | | | | | | | | | |
| DNA extraction-free | Positive | 349 | 9 | 0.063 | 0.92 | 97.5 | 95.0 | 94.6 | 97.7 |
| | Negative | 20 | 381 | | | | | | |
| **Composite (reference)**[a] | | | | | | | | | |
| Culture enrichment | Positive | 369 | 9 | 0.007 | 0.98 | 97.7 | 100 | 100 | 97.6 |
| | Negative | 0 | 381 | | | | | | |
| **Composite (reference)** | | | | | | | | | |
| DNA extraction-free | Positive | 358 | 0 | <0.0001 | 0.95 | 100 | 95.0 | 94.7 | 100 |
| | Negative | 20 | 381 | | | | | | |

[a]The composite reference was defined as pneumococcal positivity by either culture-enrichment method or the extraction-free method. PPV, positive predictive value; NPV, negative predictive value. *P*-value was derived using McNemar's test.

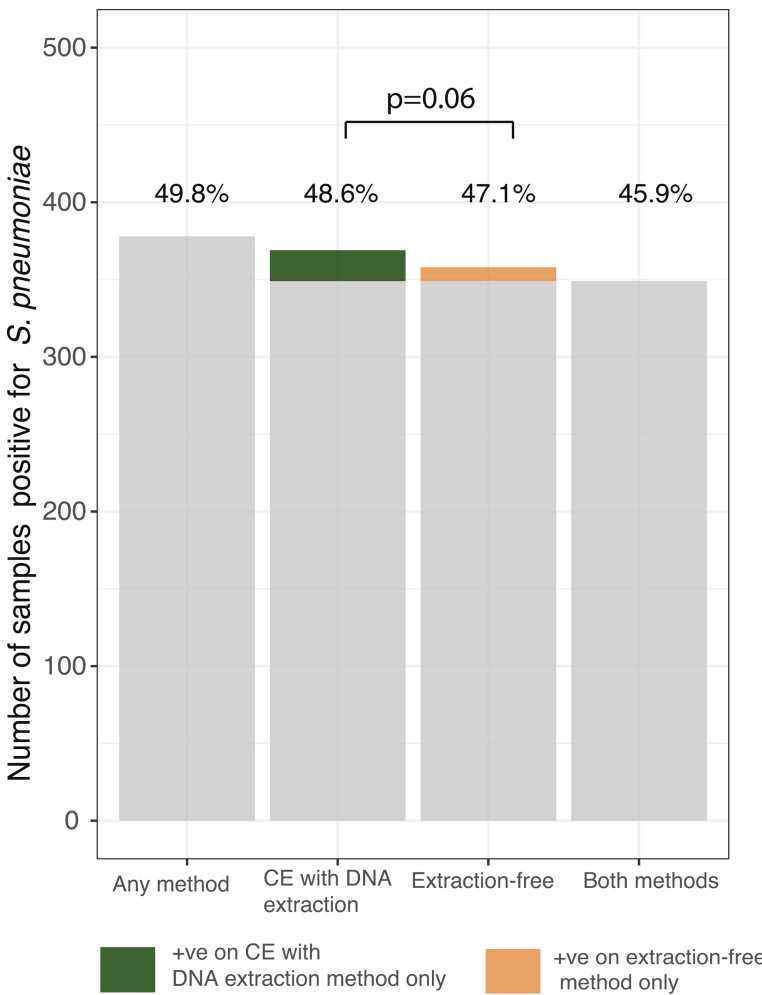

**FIG 2** Detection of pneumococcus in raw, unsupplemented saliva using methods with and without sample culture enrichment and DNA extraction. Overall, 378/759 (49.8%) saliva samples tested positive for pneumococcus (*piaB* qPCR $C_T$ value < 40) when tested by either culture enrichment followed by DNA extraction ("CE with DNA extraction") and a culture- and extraction-free method ("Extraction-free"). Of these, 369/759 (48.6%) tested positive using the CE with DNA extraction method compared to 358/759 (47.1%) when using the extraction-free method; 348/759 (45.9%) tested positive in both methods. The proportion of samples found to be positive using the CE with DNA extraction method highlighted in green denotes those that tested positive using this method only (*n* = 20). The proportion of samples found to be positive using the extraction-free method highlighted in orange denotes samples that tested positive when using this method only (*n* = 9).

by higher qPCR $C_T$ values) when using the extraction-free protocol as compared to the culture-enrichment method ($\Delta C_T = 6.68$, $P < 0.00001$; Table S3). This difference in $C_T$ values between the two methods did not vary by sampling period ($\Delta C_T = 1.26$, $P = 0.09$; Table S4).

## Reagent cost for pneumococcal carriage detection per sample

Finally, we conducted a comparison of the expenses associated with laboratory reagents, aiming to assess the cost-effectiveness of the simplified extraction-free method. Overall, the reagent expenses incurred by using saliva lysates (US$2.53/sample) were at least fivefold lower when compared to the costs incurred when extracted DNA from culture-enriched samples was used, which ranged from US$13.60 to US$19.46 depending on whether the Gent plates used were made in-house or acquired commercially in the

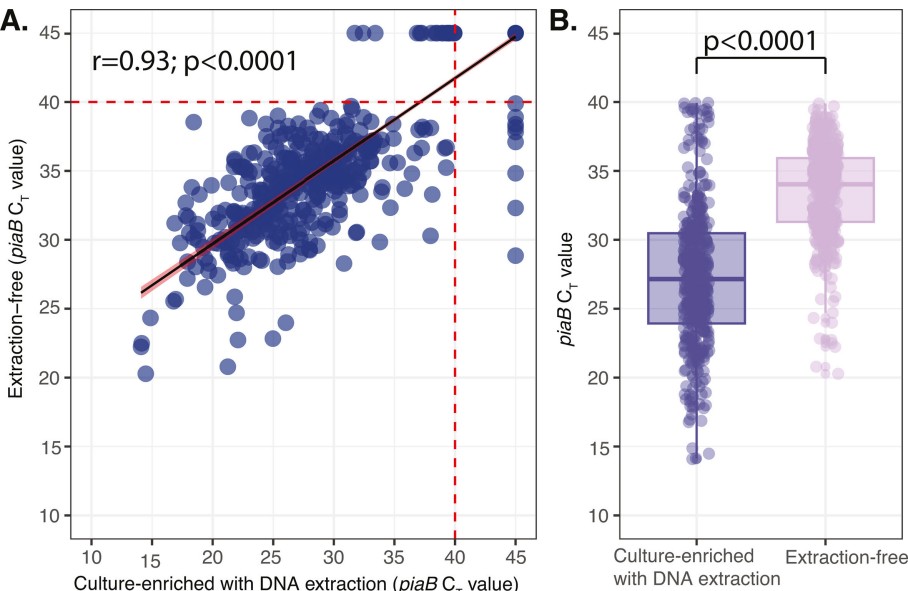

**FIG 3** Comparison of *piaB* qPCR $C_T$ values obtained when either lysates from raw saliva or DNA extracted from culture-enriched saliva samples were tested in qPCR for the presence of pneumococcus. (A) Scatter plot depicting correlation of $C_T$ values for pneumococcus-specific gene, *piaB*, obtained following testing of saliva samples processed by the culture-enrichment and DNA extraction and the extraction-free protocols. The red dotted line marks the threshold assigned to discriminate between positive and negative samples ($C_T$ value = 40). (B) Box plot of the difference in *piaB* $C_T$ values obtained when saliva samples were tested with the extraction-free method as compared to testing following sample culture enrichment and DNA extraction. The difference in $C_T$ values is depicted on the *y*-axis, and the average $C_T$ value from the samples is on the *x*-axis.

US (Table 2; Table S5 for detailed cost breakdown). The culture-enrichment method took ~24 hours from sample processing to obtaining a result as it required overnight incubation during the culture-enrichment step (Fig. 1), while the extraction-free method only required ~2 hours to obtain a result. These estimated costs do not account for general laboratory consumables, e.g., pipette tips, required equipment, and instruments, such as qPCR machines, or personnel time.

**TABLE 2** Costs of pneumococcal carriage detection in saliva samples when tested using qPCR following an extraction-free protocol or culture enrichment and DNA extraction

| Items[a] | Price per sample[b] | |
|---|---|---|
| | **Culture enrichment with DNA extraction protocol** | **Extraction-free protocol** |
| Saliva collection | $1.14 | $1.14 |
| Sample culture enrichment | $2.04–$7.90[c] | –[e] |
| MagMAX Viral/Pathogen Nucleic Acid Isolation reagents | $3.81 | $0.16[d] |
| Additional consumables for automated DNA extraction on Kingfisher Apex | $5.38 | – |
| *piaB* qPCR | $1.23 | $1.23 |
| Total cost per sample | $13.60–$19.46 | $2.53 |

[a]Specific details of reagents, suppliers/brands, and their respective catalog numbers are included in Table S5.
[b]Price per sample is calculated based on the prices listed on the vendor websites and does not include additional costs for general laboratory consumables, e.g., pipette tips, required equipment, and instruments, such as qPCR machines, or personnel time.
[c]Price range of each sample processed by culture enrichment depends on whether a commercial or in-house made Gent plate was used.
[d]Includes only the cost for proteinase K (Table S5).
[e]–, not applicable

## DISCUSSION

Oral samples have emerged as a promising sample type for the surveillance of respiratory pathogens. For the detection of pneumococcus, the recommended approach for polymicrobial samples, such as saliva, is to culture-enrich samples, preferably on blood agar plates supplemented with gentamicin, and perform a qPCR on DNA extracted from the culture-enriched samples (14). Deviating from this recommended approach could impact the sensitivity of detection, as culture enrichment increases the pneumococcal load in the sample and the DNA extraction processes concentrate the DNA through its elution into a smaller volume compared to the input volume of saliva. Nonetheless, efforts to utilize minimally processed saliva samples for pneumococcal detection to reduce sample processing time and required resources have been made (21). Having developed a saliva-based, extraction-free PCR test for SARS-CoV-2 to simplify testing during the COVID-19 pandemic, we expanded upon this protocol and evaluated its potential use for the detection of pneumococcus as a more cost-effective and sustainable option for pneumococcal carriage surveillance. Prior to this, limited evidence existed for whether complete removal of the DNA extraction and purification step would provide quality and sufficient DNA template for qPCR amplification, particularly for polymicrobial samples. Yet, remarkably, when we applied a culture- and extraction-free protocol to raw unsupplemented saliva, we obtained comparable rates of pneumococcal detection to when DNA extracted from culture-enriched saliva was tested. Moreover, the estimated cost of pneumococcal detection (based on ordering from US suppliers) using the extraction-free method was five- to sevenfold lower compared to the DNA extracted from culture-enriched saliva. This finding suggests that an extraction-free method could provide a less resource-intensive option for carriage detection as compared to the widely utilized culture-enrichment approach.

Despite the overall higher concordance in pneumococcal detection and strong $C_T$ values correlation observed between the two methods, $C_T$ values obtained from the saliva lysates were weaker compared to those obtained from culture-enriched saliva. The observed differences in the resulting $C_T$ values between the two methods ($\Delta C_T$ value = 6.69) were not unexpected. This observed reduction in sensitivity is in line with previous findings where a difference in $C_T$ values was observed when comparing DNA extracted from saliva, with and without culture enrichment (3). Collectively, these findings indicate that the observed impact on $C_T$ values in the current study primarily results from the absence of culture enrichment rather than the total omission of DNA extraction using commercial kits, which additionally offers purification and concentration of the target. Given that DNA extraction did not have a significant impact on the sensitivity of detection of pneumococcus, our findings support the possibility of omitting DNA extraction using commercial kits when testing raw saliva samples without the culture-enrichment step. With weaker $C_T$ values obtained when using the extraction-free method suggesting that the use of saliva lysates may have limitations when applied to samples with a low pneumococcal bacterial load, the extraction-free method can also be used to identify samples that can be prioritized for culture and DNA extraction, thereby drastically reducing testing costs and time required for detection.

In several instances, we detected discordant results between the extraction-free protocol and culture-enrichment method. The discrepancies were observed mostly in samples with $C_T$ values > 35 from either method. In cases where a positive result was yielded using culture-enriched saliva only, it is likely due to the higher limit of detection of the extraction-free assay, particularly in samples with lower pneumococcal load. The cause for these inconsistent results in cases where the sample tested positive by the extraction-free method only is not entirely clear but could be driven by the possible detection of residual DNA in the absence of live cells or possible growth inhibition of pneumococcus by commensal flora in saliva during the culture-enrichment step, hence yielding a positive result on the extraction-free method but being missed when saliva is culture-enriched (21). Although this result was unexpected, it highlights situations where

culture enrichment may underperform in polymicrobial samples, and, therefore, further studies are required to investigate these situations to confirm these observations.

A major limitation of our study is that we exclusively used samples collected from young children, a population known for high pneumococcal densities (22, 23). Therefore, our results cannot be generalized to other age groups, particularly the adult population, in which both overall carriage as well as carriage density are typically lower, while non-pneumococcal *Streptococcus* spp. may also be more prevalent (24). Nevertheless, high-quality saliva samples, whether collected from adults or infants, should provide the same sensitivity in samples with high pneumococcal density when standardized and clear guidelines for sample collection are provided to participants or healthcare workers (17, 25, 26). In this study, we classified samples as positive for pneumococcus using only the *piaB* qPCR assay. While *lytA* and *sp2020* assays are also commonly used (27, 28), they lack specificity for pneumococcus as compared to *piaB* as closely related non-pneumococcal *Streptococcus* spp. carry homologs of these genes (24, 28). Therefore, to evaluate our sample processing methods for their impact on pneumococcal carriage detection and to exclude confounding from other streptococci, we limited our analyses to *piaB* (24, 26). Nonetheless, the results obtained here, when testing the culture-enriched saliva samples using *piaB,* highly correlated with the results obtained when targeting the *lytA* gene in the same culture-enriched saliva samples (see Fig. S2). However, a multiplex PCR assay of the key targets used for pneumococcal carriage detection could be further evaluated. Similarly, we did not evaluate the impact of the extraction-free method on pneumococcal serotype detection; confirming its performance when combined with serotype-specific qPCR assays will be important should this method be applied in studies estimating the distribution of vaccine-type pneumococcal prevalence rates to evaluate the impact of novel vaccination strategies for pneumococcus. Finally, we estimated the costs of pneumococcal detection using US suppliers that are used in our laboratory. Therefore, these costs may change when different suppliers are used.

In conclusion, our results support the potential use of a culture- and extraction-free protocol as a cost-effective alternative to the more resource-intensive culture-enrichment-based method for pneumococcal carriage detection in children. With broad application, this extraction-free approach could significantly reduce the costs of surveillance efforts. This cost-saving benefit could support either a greater number of studies to be conducted in more regions or a large sample size collection to generate comprehensive data on pneumococcal carriage in these communities. Data generated from these surveillance efforts will provide insight into the effectiveness of the existing pneumococcal vaccines and help to inform future pneumococcal vaccine strategies. Moreover, when combined with already existing saliva-based, extraction-free assays for the detection of respiratory tract viruses, these assays can be used concurrently for the detection of viral-pneumococcal co-infection, allowing us to better understand the relationship between these pathogens and facilitate the implementation of effective clinical and public health responses.

## MATERIALS AND METHODS

### Study design

We compared methods for the detection of carriage of the pneumococcus in saliva samples collected weekly from children to screen for the prevalence of SARS-CoV-2 in New Haven childcare centers over the 2020–2021 and 2021–2022 school years. The methods used for the enrollment of study participants and at-home saliva collection have been previously described (29). After testing for SARS-CoV-2, the remaining sample volumes were stored at −80°C until further analysis. Due to sample handling requirements during the COVID-19 pandemic, for the samples collected during the 2020–2021 school year, unsupplemented raw saliva was stored and used for the detection of pneumococcus. For samples collected during the 2021–2022 school year, saliva aliquots for the detection of pneumococcus using the culture enrichment method were

supplemented with Brain Heart Infusion (BHI; Oxoid) supplemented with 10% glycerol, while the remaining aliquot tested using the extraction-free method was stored as unsupplemented raw saliva.

## Standard culture enrichment and DNA extraction protocol for pneumococcal carriage detection

Samples were thawed on ice, and 100 µL of raw saliva was inoculated on trypticase soy agar (TSA II) supplemented with 5% defibrinated sheep blood and 10 mg/L gentamicin (Gent plates, ThermoFisher Scientific) and incubated overnight at 37°C with 5% $CO_2$ (26). Bacterial growth was harvested into 2,100 µL of BHI supplemented with 10% glycerol and stored at −80°C until further processing. These samples were considered culture-enriched for pneumococcus. DNA was extracted from 200 µL of culture-enriched samples using the MagMAX Viral/Pathogen Nucleic Acid Isolation kit (ThermoFisher Scientific), following a modified protocol on the KingFisher Apex (ThermoFisher Scientific) as previously described (13).

## Extraction-free protocol

Saliva samples were thawed on ice, and lysates were prepared following the "SalivaDirect" extraction-free protocol as previously described (Fig. 1) (17). Briefly, a total of 6.5 µL (20 mg/mL) of proteinase K was added to 50 µL of each saliva sample, aliquoted into strip tubes. The tubes were firmly sealed, placed in a rack, and vortexed for 1 minute at 3,200 rpm. Samples were then heated on a thermocycler at 95°C for 5 minutes.

## Molecular detection of pneumococcal carriage

A real-time qPCR assay targeting the pneumococcal iron uptake ABC transporter lipoprotein *piaB* gene was used to detect the presence of pneumococcal DNA in both the DNA extracted from culture-enriched samples and saliva lysates, as previously described (12, 13, 26). Both DNA extracts and saliva lysates (2.5 µL) were tested in 20 µL reaction volumes using Advanced Universal Probe Supermix (BioRad, USA) and a primer/probe mix at concentrations of 250 nM (1 µL per reaction) (Iowa Black quenchers, USA). A standard curve prepared from five 1:10 serial dilutions of pneumococcal DNA, ranging from 1 to 0.00001 ng/µL was included as a positive control. Water was added as a non-template/negative qPCR control in each run. In qPCR runs where extracted DNA was used as a template, a negative extraction control (BHI + 10% glycerol) was included. Samples were classified as positive for pneumococcus when a *piaB* cycle threshold ($C_T$) value was <40 (13, 26).

## Limit of assay detection for the extraction-free approach and culture-enriched approach

Saliva samples were collected from healthy volunteers ($n$ = 5) and screened for the absence of pneumococcus using qPCR targeting *piaB*, as previously described (16). Samples negative for *piaB* were considered negative for pneumococcus and were pooled together. A serotype 19A pneumococcal strain was serially diluted 1:10 into the pooled saliva, from $5 \times 10^7$ to $5 \times 10^1$ CFU/mL (Fig. S1A) (30). Each dilution was tested in triplicate, using both the extraction-free method and culture-enrichment methods as described above. The limit of detection was determined as the bacterial concentration in which all the triplicates were positive for pneumococcus ($C_T$ < 40). Because the limit of detection for the culture-enriched approach was not determined at the lowest 1:10 dilution ($5 \times 10^1$ CFU/mL), the limit of detection for the culture-enriched approach was further performed using 1:2 serial dilution, from 500 to 8 CFU/mL. For both assays, unspiked saliva was included as a negative control (Fig. S1A).

## Statistical analysis

Cohen's kappa was used to assess the chance-corrected agreement of identifying pneumococcal carriage based on the two methods (31). Cohen's kappa ratios of 0, 0.01–0.20, 0.21–0.40, 0.41–0.60, 0.61–0.80, and >0.81 were interpreted as exhibiting poor, slight, fair, moderate, substantial, and near-perfect agreement, respectively (31, 32). In addition, McNemar's test was used to assess the detection disagreement between the two methods and complement the obtained Cohen's Kappa agreement. Screening test parameters (sensitivity, specificity, positive and negative predictive values) were calculated using the "caret" R package by using the culture-enrichment protocol or composite reference as the "reference standard," where pneumococcal positivity was defined as being positive by either culture-enrichment method or the extraction-free method (33). We used linear regression to evaluate the differences in $C_T$ values obtained using the two methods. Because samples collected between the two sampling periods were stored differently (i.e., without supplementation in the 2020–2021 school year and supplemented with BHI + 10% glycerol in the 2021–2022 school year) before culture enrichment was performed, we fitted a linear regression model adjusting for the sampling period to take into account the differences in storage conditions between the two sampling periods. An interaction term between the method and sampling period was also fitted to evaluate whether the effect of the method varied by sampling period. *P*-values < 0.05 were considered significant. All analyses were performed using R (version 4.3.0).

## ACKNOWLEDGMENTS

We thank the study participants and their families for their time and dedication to our study. We thank all members of the Trackcare study team, Yale Pathology Labs, and the Wyllie Lab at the Yale School of Public Health for their critical contributions to the initial research study from which these samples were obtained.

The study was supported by a Fast grant from Emergent Ventures at the Mercatus Center at George Mason University (A.L.W.) and research funds and salary support from SalivaDirect, Inc. (A.L.W.). The study protocol was designed by the Yale researchers. The decision to publish was made by the Yale researchers; all authors agreed with the decision to publish and with the results of the study.

C.P., A.L.W., and O.M.A. conceptualized the study. A.K.B. and A.L.W. managed the collection of samples. T.-Y.L., D.Y.-C., K.F., and O.M.A. developed and validated the laboratory methods. C.P., T.-Y.L., M.S.H., D.Y.-C., K.F., and A.Y. conducted laboratory experiments. C.P. and T.-Y.L. analyzed the data. D.M.W. and V.E.P. provided statistical advice. C.P., T.-Y.L., and A.L.W. drafted the manuscript. All authors amended, critically reviewed, and commented on the final manuscript.

## AUTHOR AFFILIATION

[1]Department of Epidemiology of Microbial Diseases, Yale School of Public Health, New Haven, Connecticut, USA

## AUTHOR ORCIDs

Chikondi Peno http://orcid.org/0000-0003-0083-119X
Anna York http://orcid.org/0000-0002-1332-8983
Orchid M. Allicock http://orcid.org/0000-0002-6570-5453
Anne L. Wyllie http://orcid.org/0000-0001-6015-0279

## AUTHOR CONTRIBUTIONS

Tzu-Yi Lin, Formal analysis, Investigation, Writing – original draft, Writing – review and editing | Maikel S. Hislop, Investigation, Writing – review and editing | Devyn Yolda-Carr, Investigation, Methodology, Validation, Writing – review and editing | Katherine Farjado,

Investigation, Writing – review and editing | Anna York, Investigation, Methodology, Writing – review and editing | Virginia E. Pitzer, Formal analysis, Writing – review and editing | Daniel M. Weinberger, Formal analysis, Methodology, Writing – review and editing | Amy K. Bei, Funding acquisition, Resources, Writing – review and editing | Orchid M. Allicock, Conceptualization, Investigation, Methodology, Validation, Writing – review and editing | Anne L. Wyllie, Conceptualization, Funding acquisition, Investigation, Methodology, Resources, Supervision, Writing – original draft, Writing – review and editing.

## DATA AVAILABILITY

The data used to support the findings presented in this study are available and will be submitted upon reasonable request to the corresponding authors.

## ETHICS APPROVAL

The collection of saliva samples from the children in childcare centers (New Haven, CT, USA) was approved by the Institutional Review Board of the Yale Human Research Protection Program (protocol number 200002839). Written informed consent was obtained from the parent or guardian of every participating child. The collection of de-identified saliva samples from healthy volunteers for assay validation was approved by the Institutional Review Board of the Yale Human Research Protection Program (protocol number 2000029374).

## ADDITIONAL FILES

The following material is available online.

### Supplemental Material

**Supplemental material (Spectrum00591-24-s0001.docx).** Tables S1-S5; Fig. S1 and S2.

### Open Peer Review

**PEER REVIEW HISTORY (review-history.pdf).** An accounting of the reviewer comments and feedback.

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
