## [Reviewer comments · Microbiology Spectrum]

Microbiology Spectrum

A low-cost culture- and DNA extraction-free method for the molecular detection of pneumococcal carriage in saliva

Chikondi Peno, Tzu -Yi Lin, Maikel Hislop, Devyn Yolda-Carr, Katherine Fajardo, Anna York, Virginia Pitzer, Daniel Weinberger, Amy Bei, Orchid Allcock, and Anne Wyllie

Corresponding Author(s): Chikondi Peno, Yale University School of Public Health

Review Timeline:

Submission Date:	March 6, 2024
Editorial Decision:	April 15, 2024
Revision Received:	April 26, 2024
Accepted:	May 5, 2024

Editor: Wendy Szymczak

Reviewer(s): Disclosure of reviewer identity is with reference to reviewer comments included in decision letter(s). The following individuals involved in review of your submission have agreed to reveal their identity: Annalisa Pantosti (Reviewer #1)

Transaction Report:

DOI: <https://doi.org/10.1128/spectrum.00591-24>

Re: Spectrum00591-24 (A low-cost culture- and DNA extraction-free method for the molecular detection of pneumococcal carriage in saliva)

Dear Dr. Chikondi Peno:

Thank you for the privilege of reviewing your work. Below you will find my comments, instructions from the Spectrum editorial office, and the reviewer comments.

Revision Guidelines

Sincerely,
Wendy Szymczak
Editor
Microbiology Spectrum

Reviewer #1 (Comments for the Author):

The paper by C. Peno and coworkers describes a simplified qPCR method to detect pneumococcal carriage in saliva of children. The paper is well-written and straight-forward. The proposed method is easy and economical both in terms of time and money. In the children population studied it reaches good agreement with the established method requiring culture enrichment, although it is obviously less sensitive with a higher detection limit. The limitations of the method and the limitations of the present study

are clearly discussed by the authors.

Reviewer #2 (Comments for the Author):

The authors present A low-cost culture- and DNA extraction-free method for the molecular detection of pneumococcal carriage in saliva. Overall, this study is well done and there is really no major issue concerning the methodology. However, I have some comments.

- Both of the methods are quite similar (piaB qPCR). Although the authors explain well, I would consider removing "gold standard". Perhaps use 'reference standard'?
- It is great to see that amplification from saliva works. I would have expected some inhibition but this does not seem to be the case. However, saliva of kids might be 'liquid' and therefore easy (easier) to pipette as compared to saliva from adults? I guess it is important that the texture of the saliva samples is homogenous? However, is this easy to standardize? Could the authors comment?
- I am a bit surprised about the estimated costs of pneumococcal carriage. Sample culture-enrichment seems quite high for me. Do I understand correctly, that this is the price for a single Gent plate (either in-house or commercial made)?
- This goes beyond the actual manuscript: I appreciate that the authors tried to characterize actual isolates in an earlier publication. I agree that this is a difficult work due to the 'background noise' of other bacterial species. However, other approaches (e.g. sequencing with prior whole genome amplification) could be considered to further validate the piaB qPCR methods in saliva samples in the future.

Reviewer #1

- The paper by C. Peno and coworkers describes a simplified qPCR method to detect pneumococcal carriage in saliva of children. The paper is well-written and straight-forward. The proposed method is easy and economical both in terms of time and money. In the children population studied it reaches good agreement with the established method requiring culture enrichment, although it is obviously less sensitive with a higher detection limit. The limitations of the method and the limitations of the present study are clearly discussed by the authors.

We thank the reviewer for taking the time to review our work and for the kind remarks.

Reviewer #2

- The authors present A low-cost culture- and DNA extraction-free method for the molecular detection of pneumococcal carriage in saliva. Overall, this study is well done and there is really no major issue concerning the methodology. However, I have some comments.

We thank the reviewer for the feedback and hope our edits to the paper have clarified and/or improved upon the points raised by the reviewer.

- Both of the methods are quite similar (piaB qPCR). Although the authors explain well, I would consider removing "gold standard". Perhaps use 'reference standard'?

We agree with the reviewer, we have now revised the manuscript and now refer to the culture - enrichment method as a "reference standard".

- It is great to see that amplification from saliva works. I would have expected some inhibition but this does not seem to be the case. However, saliva of kids might be 'liquid' and therefore easy (easier) to pipette as compared to saliva from adults? I guess it is important that the texture of the saliva samples is homogenous? However, is this easy to standardize? Could the authors comment?

Through our work validating and optimizing saliva as a sample type for the detection of respiratory pathogens, we agree with the reviewer that we – and others – have had concerns that saliva may have inhibitors that may interfere with PCR. When using this approach in our RNA extraction-free test for SARS-CoV-2 that we received US FDA emergency use authorization for, the FDA required us to test for interfering substances. The only interference that we found was from mucus/white blood cells (which you could get if sputum was collected) or very high concentrations of Colgate toothpaste. For this reason, we developed very clear collection instructions and sample accessioning instructions that require us to visually inspect the samples that are received to confirm that they are indeed true saliva (instructions also outlined here [1]). Millions of saliva samples have now been tested using the SARS-CoV-2 assay, in which we also target human RNase P as a sample quality control and we have observed an incredibly low fraction of samples reported with interference (<0.05%). In other settings, the variability in sensitivity of saliva as a sample type is mostly experienced due to lack of standardized saliva collection methods[2].

Due to this experience and the provision of clear instructions on the proper collection of samples, combined with the fact that these samples were previously tested in the SARS-CoV-2 assay so we know the status of the detection of human RNase P [1], we are confident that we did not experience high levels of inhibition in our extraction-free assay for pneumococcus.

In regards to the consistency, we again find that the sample collection instructions that we provide help us to receive easy to pipette samples from individuals of all ages, for example in our

previously reported National Basketball Association (NBA) cohort [3], in children [1], including older adults (>60 years of age)[4].

We have now discussed this in line 215-217 of the manuscript

- I am a bit surprised about the estimated costs of pneumococcal carriage. Sample culture-enrichment seems quite high for me. Do I understand correctly that this is the price for a single Gent plate (either in-house or commercial made)?

Indeed, the culture-enrichment process cost may seem higher for a single sample, this is because the cost of culture-enrichment method reported in our manuscript includes a kit-based DNA extraction costs which is the most expensive part of this process (\$3.81 for manual extraction, and \$9.19 for automated extraction, Supplementary Table 5). We estimated the cost of culture-enrichment without DNA extraction to be \$2.04 when in-house Gent plates are used vs. \$7.90 when commercial plates from Remel are used (Supplementary Table 5). However, we note that our costs are estimated using suppliers that we use in our lab. Therefore, these costs maybe lower if different suppliers are used for estimations. We apologize to the reviewer for the confusion surrounding this and have clarified this in the results (line 153-154) and in the discussion (line 178, 231-233).

- This goes beyond the actual manuscript: I appreciate that the authors tried to characterize actual isolates in an earlier publication. I agree that this is a difficult work due to the 'background noise' of other bacterial species. However, other approaches (e.g. sequencing with prior whole genome amplification) could be considered to further validate the *piaB* qPCR methods in saliva samples in the future.

We thank the reviewer for thinking on this point with us. We are indeed exploring ways to demonstrate the presence of pneumococci detected in saliva by qPCR, such as our attempt at using magnetic bead separation method [5]. We have also tested saliva samples in assays for *lytA* gene and some also in *SP2020* gene to explore concordance with *piaB* gene. We do observe a high degree of specificity for *piaB*, which we continue to find reassuring, but indeed we continue to be aware and cautious of [6, 7].

References

1. Rayack, E.J., et al., *Routine saliva testing for SARS-CoV-2 in children: Methods for partnering with community childcare centers*. Front Public Health, 2023. **11**: p. 1003158.
2. Tan, S.H., et al., *Saliva as a gold-standard sample for SARS-CoV-2 detection*. Lancet Respir Med, 2021. **9**(6): p. 562-564.
3. Vogels, C.B.F., et al., *SalivaDirect: A simplified and flexible platform to enhance SARS-CoV-2 testing capacity*. Med, 2021. **2**(3): p. 263-280.e6.
4. Wyllie, A.L., et al., *Persistence of Pneumococcal Carriage among Older Adults in the Community despite COVID-19 Mitigation Measures*. Microbiol Spectr, 2023. **11**(3): p. e0487922.
5. York, A., et al., *Magnetic bead-based separation of pneumococcal serotypes*. Cell Rep Methods, 2023. **3**(2): p. 100410.
6. Hislop, M.S., et al., *High Levels of Detection of Nonpneumococcal Species of Streptococcus in Saliva from Adults in the United States*. Microbiol Spectr, 2023. **11**(3): p. e0520722.

7. Wyllie, A.L., et al., *Molecular surveillance of nasopharyngeal carriage of Streptococcus pneumoniae in children vaccinated with conjugated polysaccharide pneumococcal vaccines*. Sci Rep, 2016. **6**: p. 23809.

Re: Spectrum00591-24R1 (A low-cost culture- and DNA extraction-free method for the molecular detection of pneumococcal carriage in saliva)

Dear Dr. Chikondi Peno:

Your manuscript has been accepted, and I am forwarding it to the ASM production staff for publication. Your paper will first be checked to make sure all elements meet the technical requirements. ASM staff will contact you if anything needs to be revised before copyediting and production can begin. Otherwise, you will be notified when your proofs are ready to be viewed.

Sincerely,
Wendy Szymczak
Editor
Microbiology Spectrum